# STRENGTH-ADAPTIVE ADVERSARIAL TRAINING

## ABSTRACT

Adversarial training (AT) is proved to reliably improve network's robustness against adversarial data. However, current AT with a pre-specified perturbation budget has limitations in learning a robust network. Firstly, applying a pre-specified perturbation budget on networks of various model capacities will yield divergent degree of robustness disparity between natural and robust accuracies, which deviates from robust network's desideratum. Secondly, the attack strength of adversarial training data constrained by the pre-specified perturbation budget fails to upgrade as the growth of network robustness, which leads to robust over-fitting and further degrades the adversarial robustness. To overcome these limitations, we propose *Strength-Adaptive Adversarial Training* (SAAT). Specifically, the adversary employs an adversarial loss constraint to generate adversarial training data. Under this constraint, the perturbation budget will be adaptively adjusted according to the training state of adversarial data, which can effectively avoid robust overfitting. Besides, SAAT explicitly constrains the attack strength of training data through the adversarial loss, which manipulates model capacity scheduling during training, and thereby can flexibly control the degree of robustness disparity and adjust the tradeoff between natural accuracy and robustness. Extensive experiments show that our proposal boosts the robustness of adversarial training.

## 1 INTRODUCTION

Current deep neural networks (DNNs) achieve impressive breakthroughs on a variety of fields such as computer vision (He et al., 2016), speech recognition (Wang et al., 2017), and NLP (Devlin et al., 2018), but it is well-known that DNNs are vulnerable to adversarial data: small perturbations of the input which are imperceptible to humans will cause wrong outputs (Szegedy et al., 2013; Goodfellow et al., 2014). As countermeasures against adversarial data, adversarial training (AT) is a method for hardening networks against adversarial attacks (Madry et al., 2017). AT trains the network using adversarial data that are constrained by a pre-specified perturbation budget, which aims to obtain the output network with the minimum adversarial risk of an sample to be wrongly classified under the same perturbation budget. Across existing defense techniques, AT has been proved to be one of the most effective and reliable methods against adversarial attacks (Athalye et al., 2018).

Although promising to improve the network's robustness, AT with a pre-specified perturbation budget still has limitations in learning a robust network. Firstly, the pre-specified perturbation budget is inadaptable for networks of various model capacities, yielding divergent degree of robustness disparity between natural and robust accuracies, which deviates from robust network's desideratum. Ideally, for a robust network, perturbing the attack budget within a small range should not cause signifcant accuracy degradation. Unfortunately, the degree of robustness disparity is intractable for AT with a pre-specified perturbation budget. In standard AT, there could be a prominent degree of robustness disparity in output networks. For instance, a standard PGD adversarially-trained PreAct ResNet18 network has 84% natural accuracy and only 46% robust accuracy on CIFAR10 under $\ell_\infty$ threat model, as shown in Figure 1(a). Empirically, we have to increase the pre-specified perturbation budget to allocate more model capacity for defense against adversarial attacks to mitigate the degree of robustness disparity, as shown in Figure 1(b). However, the feasible range of perturbation budget is different for networks with different model capacities. For example, AT with perturbation budget $\epsilon = 40/255$ will make PreAct ResNet-18 optimization collapse, while wide ResNet-34-10 can learn normally. In order to maintain a steady degree of robustness disparity, we have to find *separate* perturbation budgets for *each* network with different model capacities. Therefore, it may be pessimistic to use AT with a pre-specified perturbation budget to learn a robust network.

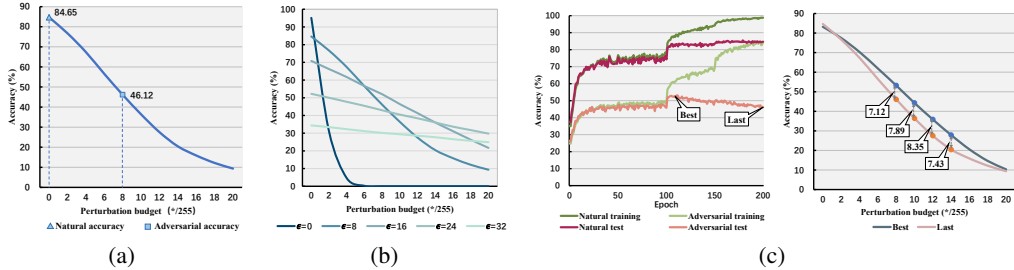

Figure 1: Robustness evaluation on different test perturbation budgets of (a) standard AT; (b) AT with different training pre-specified perturbation budgets. (c) The learning curve of standard AT with pre-specified perturbation $\epsilon = 8/255$ on PreAct ResNet-18 under $\ell_\infty$ threat model and the robustness evaluation of its "best" and "last" checkpoints.

Secondly, the attack strength of adversarial training data constrained by the pre-specified perturbation budget is gradually weakened as the growth of network robustness. During the training process, adversarial training data are generated on the fly and are changed based on the updating of the network. As the the network's adversarial robustness continues to increase, the attack strength of adversarial training data with the pre-specified perturbation budget is getting relatively weaker. Given the limited network capacity, a degenerate or stagnant adversary accompanied by an evolving network will easily cause training bias: adversarial training is more inclined to the defense against weak strength attacks, and thereby erodes defenses on strong strength attacks, leading to the undesirable robust overfitting, as shown in Figure 1(c). Moreover, compared with the "best" checkpoint in AT with robust overfitting, the "last" checkpoint's defense advantage in weak strength attack is slight, while its defense disadvantage in strong strength attack is significant, which indicates that robust overfitting not only *exacerbates* the degree of robustness disparity, but also further *degrades* the adversarial robustness. Thus, it may be deficient to use adversarial data with a pre-specified perturbation budget to train a robust network.

To overcome these limitations, we propose strength-adaptive adversarial training (SAAT), which employs an adversarial loss constraint to generate adversarial training data. The adversarial perturbation generated under this constraint is adaptive to the dynamic training schedule and networks of various model capacities. Specifically, as adversarial training progresses, a larger perturbation budget is required to satisfy the adversarial loss constraint since the network becomes more robust. Thus, the perturbation budgets in our SAAT is adaptively adjusted according to the training state of adversarial data, which restrains the training bias and effectively avoids robust overfitting. Besides, SAAT explicitly constrains the attack strength of training data by the adversarial loss constraint, which guides model capacity scheduling in adversarial training, and thereby can flexibly adjust the tradeoff between natural accuracy and robustness, ensuring that the output network maintains a steady degree of robustness disparity even under networks with different model capacities.

Our contributions are as follows. **(a)** In standard AT, we characterize the pessimism of adversary with a pre-specified perturbation budget, which is due to the intractable robustness disparity and undesirable robust overfitting (in Section 3.1). **(b)** We propose a new adversarial training method, i.e., SAAT (its learning objective in Section 3.2 and its realization in Section 3.3). SAAT is a general adversarial training method that can be easily converted to natural training or standard AT. **(c)** Empirically, we find that adversarial training loss is well-correlated with the degree of robustness disparity and robust generalization gap (in Section 4.2), which enables our SAAT to overcome the issue of robust overfitting and flexibly adjust the tradeoff of adversarial training, leading to the improved natural accuracy and robustness (in Section 4.3).

## 2 PRELIMINARY AND RELATED WORK

In this section, we review the adversarial training method and related works.

### 2.1 ADVERSARIAL TRAINING

**Learning objective.** Let $f_\theta$, $\mathcal{X}$ and $\ell$ be the network $f$ with trainable model parameter $\theta$, input feature space, and loss function, respectively. Given a $C$-class dataset $\mathcal{S} = \{(x_i, y_i)\}_{i=1}^n$, where $x_i \in \mathcal{X}$ and $y_i \in \mathcal{Y} = \{0, 1, ..., C-1\}$ as its associated label. In natural training, most machine

learning tasks could be formulated as solving the following optimization problem:

$$\min_{\theta} \frac{1}{n} \sum_{i=1}^{n} \ell(f_{\theta}(x_i), y_i). \tag{1}$$

The learning objective of natural training is to obtain the networks that have the minimum empirical risk of a natural input to be wrongly classified. In adversarial training, the adversary adds the adversarial perturbation to each sample, i.e., transform $\mathcal{S} = \{(x_i, y_i)\}_{i=1}^{n}$ to $\mathcal{S}' = \{(x_i' = x_i + \delta_i, y_i)\}_{i=1}^{n}$. The adversarial perturbation $\{\delta_i\}_{i=1}^{n}$ are constrained by a pre-specified budget, *i.e.* $\{\delta \in \Delta : ||\delta||_p \leq \epsilon\}$, where $p$ can be $1, 2, \infty$, etc. In order to defend such attack, standard adversarial training (AT) (Madry et al., 2017) resort to solve the following objective function:

$$\min_{\theta} \frac{1}{n} \sum_{i=1}^{n} \max_{\delta_i \in \Delta} \ell(f_{\theta}(x_i + \delta_i), y_i). \tag{2}$$

Note that the outer minimization remains the same as Eq.(1), and the inner maximization operator can also be re-written as

$$\delta_i = \arg\max_{\delta_i \in \Delta} \ell(f_{\theta}(x_i + \delta_i), y_i), \tag{3}$$

where $x_i' = x_i + \delta_i$ is the most adversarial data within the perturbation budget $\Delta$. Standard AT employs the most adversarial data generated according to Eq.(3) for updating the current model. The learning objective of standard AT is to obtain the networks that have the minimum adversarial risk of a input to be wrongly classified under the pre-specified perturbation budget.

**Realizations.** The objective functions of standard AT (Eq.(2)) is a composition of an inner maximization problem and an outer minimization problem, with one step generating adversarial data and one step minimizing loss on the generated adversarial data w.r.t. the model parameters $\theta$. For the outer minimization problem, Stochastic Gradient Descent (SGD) (Bottou, 1999) and its variants are widely used to optimize the model parameters (Rice et al., 2020). For the inner maximization problem, the Projected Gradient Descent (PGD) (Madry et al., 2017) is the most common approximation method for generating adversarial perturbation, which can be viewed as a multi-step variant of Fast Gradient Sign Method (FGSM) (Goodfellow et al., 2014). Given normal example $x \in \mathcal{X}$ and step size $\alpha > 0$, PGD works as follows:

$$\delta^{k+1} = \Pi_{\Delta}(\alpha \cdot \text{sign}\nabla_x \ell(f(x + \delta^k), y) + \delta^k), k \in \mathbb{N}, \tag{4}$$

where $\delta^k$ is adversarial perturbation at step k; and $\Pi_{\Delta}$ is the projection function that project the adversarial perturbation back into the pre-specified budget $\Delta$ if necessary.

## 2.2 RELATED WORK

**Stopping criteria.** There are different stopping criteria for PGD-based adversarial training. For example, standard AT (Madry et al., 2017) employs a fixed number of iterations $K$, namely PGD-K, which is commonly used in many outstanding adversarial training variants, such as TRADES (Zhang et al., 2019), MART (Wang et al., 2019b), and RST (Carmon et al., 2019). Besides, some works have further enhanced the PGD-K method by incorporating additional optimization mechanisms, such as curriculum learning (Cai et al., 2018), FOSC (Wang et al., 2019a), and geometry reweighting (Zhang et al., 2020b). On the other hand, some works adopt different PGD stopping criterion, i.e., misclassification-aware criterion, which stops the iterations once the network misclassifies the adversarial data. This misclassification-aware criterion is widely used in the emerging adversarial training variants, such as FAT (Zhang et al., 2020a), MMA (Ding et al., 2018), IAAT (Balaji et al., 2019), ATES (Sitawarin et al., 2020), and Customized AT (Cheng et al., 2020). Different from these works, we propose strength-adaptive PGD (SA-PGD) that uses the minimum adversarial loss as the stopping criterion to generate efficient adversarial data for adversarial training.

**Relationship between accuracy and robustness.** Also relevant to this work are works that study the relationship between natural accuracy and robustness. PGD-based AT can enhance the robustness against adversarial data, but degrades the accuracy on the natural data significantly. One popular point is the inevitable tradeoff between robustness and natural accuracy. For example, Tsipras et al. (2018) claimed robustness and natural accuracy might at odds. Su et al. (2018) concluded a linearly negative correlation between the logarithm of natural accuracy and robustness. Zhang et al.

(2019) theoretically characterized the tradeoff. However, human is a network that is both robust and accurate with no tradeoff according to the definition of adversarial perturbation. Some other works also provide evidence that robustness and natural accuracy are not opposing. For example, Stutz et al. (2019) confirmed the existence of adversarial data on the manifold of natural data. Yang et al. (2020) showed benchmark datasets with adversarial perturbation are distributionally separated. Raghunathan et al. (2020) stated that additional unlabeled data help mitigate the tradeoff. Nakkiran (2019) proved that the tradeoff is due to the insufficient network expression ability. A separate but related line of works also has challenged the tradeoff by improving the natural accuracy while maintaining the robustness (Zhang et al., 2020a) or retaining the natural accuracy while improving the robustness (Zhang et al., 2020b). However, these works use PGD as the robustness evaluation, which is not always successful since these networks can be defeated by stronger attacks (Liu et al., 2021). In this work, we combine AutoAttack (Croce & Hein, 2020b), a stronger and more reliable robustness evaluation method, to conduct a more comprehensive evaluation of AT's tradeoff.

## 3    STRENGTH-ADAPTIVE ADVERSARIAL TRAINING

In this section, we introduce the proposed strength-adaptive adversarial training (SAAT) and its learning objective as well as algorithmic realization.

### 3.1    MOTIVATIONS OF SAAT

**Robustness disparity is intractable for AT with a pre-specified perturbation budget.**  For adversarially-trained networks, it is widely recognized that robust accuracy should be lower than the natural accuracy. Nevertheless, the degree of robustness disparity between natural and robust accuracy is often overlooked. Ideally, for a fully robust network, the robust accuracy and natural accuracy should be very close. The technique of maintaining a steady degree of robustness disparity is therefore critical for learning a robust network. However, current AT methods typically employ a pre-specified perturbation budget to generate adversarial data, whose attack strength is relative to the network's model capacity, which fails to yield a steady degree of robustness disparity across different networks, as shown in Figure 2(a). For each network, we further numerically compute their degree of robustness disparity (RD): $RD = \frac{1}{n} \sum_{i=1}^{n} \frac{A_0 - A_i}{\epsilon_i}$, where $A_i$ represents the accuracy on the perturbation budget $\epsilon_i$. As shown by the statistical results in Figure 2(b), when the perturbation budget is fixed, the degree of robustness disparity becomes more prominent as the model capacity increases, which is not robust network's desideratum. Thus, to maintain a steady degree of robustness disparity in adversarial training, we should open up perturbation budget constraints and flexibly adjust the attack strength of training data according to the network's model capacity.

**Robust overfitting degrades the network's adversarial robustness.** AT employs the most adversarial data to reduce the sensitivity of the network's output w.r.t. adversarial perturbation of the natural data. However, during the training process, adversarial training data is generated on the fly and is getting weaker for networks with increasing adversarial robustness. Given a certain amount of allocatable model capacity, the adversarial training data with a pre-specified perturbation budget will inevitably induce training bias, which eventually leads to the robust overfitting. As shown in Figure 2(c), it is observed that when the perturbation budget is fixed, robust overfitting occurs as the network's model capacity increases. We further compare the adversarial robustness between the "best" and "last" checkpoints in AT without robust overfitting and with robust overfitting. As shown in Figure 2(d), it can be seen that robust overfitting significantly degrades the network's adversarial robustness. Therefore, to avoid robust overfitting in adversarial training, we should flexibly adjust the attack strength of adversarial training data to adapt to the dynamic training schedule.

### 3.2    LEARNING OBJECTIVE OF SAAT

Let $\rho$ be the specified minimum adversarial loss constraint to adversarial training data. In the learning objective of SAAT, the outer minimization for optimizing model parameters still follows Eq.(2) or Eq.(1). However, instead of generating adversarial perturbation $\delta$ via inner maximization, we generate $\delta$ as follows:

$$\delta_i = \arg\min_{\delta_i} \ell(f_\theta(x_i + \delta_i), y_i) \quad \text{s.t.} \quad \ell(f_\theta(x_i + \delta_i), y_i) \geq \rho. \tag{5}$$

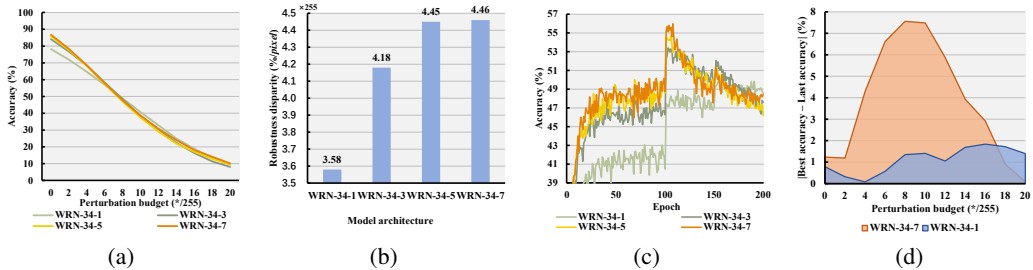

Figure 2: (a) Robustness evaluation, (b) robustness disparity and (c) testing curves of AT with perturbation budget $\epsilon = 8/255$ under $\ell_\infty$ threat model on networks with different model capacities; (d) robustness differences between the "best" and "last" checkpoints in AT without robust overfitting (WRN-34-1) and with robust overfitting (WRN-34-7).

Note that the operator $\arg\max$ in Eq.(3) is replaced with $\arg\min$ here, and there is no explicit perturbation budget constraint for $\delta$. Instead, we adopt the magnitude of adversarial loss to constrain the generation of adversarial perturbation. The constraint ensures that the loss of adversarial training data is greater than the specified minimum adversarial loss $\rho$. Among all such $\delta$ satisfying the constraint, we select the one minimizing $\ell(f_\theta(x_i + \delta_i), y_i)$. In terms of the process of generating adversarial perturbations, Eq.(5) could be regarded as an adaptive adversary, since adversarial loss is related to the training schedule and network's model capacity. For example, in the early stages of training, adversarial attack may be needless to generate qualified adversarial training data. However, in the later stages of training, more effort is required to generate the corresponding adversarial training data because the network is more robust. On the other hand, in terms of attack strength of training data, Eq.(5) is actually an attack strength-fixed adversary, since the adversarial loss of training data is constrained by a fixed $\rho$.

The learning objective of SAAT is to obtain the networks with a steady degree of robustness disparity, which is achieved by optimizing model using adversarial training data with a fixed attack strength (in terms of the adversarial loss $\rho$). $\rho$ is used to guide model capacity scheduling during adversarial training, so as to ensure that the output network maintains a steady degree of robustness disparity.

**Relation with natural training and standard AT.** Notice that the learning objective of SAAT is extremely general. When $\rho \le 0$, SAAT is equivalent to natural training (refer to Eq.(1), then all training data does not need adversarial perturbations, so that most of the model capacity will be used to learn natural data. When $\rho \to \infty$, SAAT is equivalent to standard AT (refer to Eq.(2), then all training data is the most adversarial data, so that a large amount of model capacity will be used for enhancing adversarial robustness (depending on the maximal perturbation budget). When $0 < \rho < \infty$, SAAT lies in the middle of natural training and standard AT, which can manipulate the model capacity scheduling during the training phase, so as to obtain the output network with multiple alternative forms of adversarial robustness for various practical needs. Eq.(5) recovers both natural training and standard AT, thus it is a more general learning objective of adversarial training.

### 3.3 REALIZATION OF SAAT

The learning objective of SAAT implies the optimization of an adversarially robust networks with a steady degree of robustness disparity, with one step generating qualified adversarial training data and one step minimizing the adversarial loss w.r.t. the model parameters. Specifically, we search for qualified adversarial training data by adjusting the perturbation budget and attack step. For instance, given a perturbation budget, we perform sufficient PGD attacks within this budget. If the most adversarial data still does not satisfy the constraint of Eq.(5), we will increase the perturbation budget and conduct further PGD attacks until we find adversarial data that satisfies the minimum adversarial loss criterion.

How to estimate the optimal perturbation budget is an open question. Here we heuristically design a simple implementation: progressive search. Specifically, the perturbation budgets of training data are initially set to 0, and then their perturbation budgets are increased stepwise (e.g., increments of step size $\tau$). Each time the perturbation budgets are updated, it is followed by $K$ iterations of PGD attacks until the generated adversarial data satisfies the minimum adversarial loss criterion. The limitation of this implementation is that setting the initial perturbation budget to 0 increases the

---

**Algorithm 1** Strength-Adaptive PGD (SA-PGD)

---

1: **Input:** data $x \in \mathcal{X}$, label $y \in \mathcal{Y}$, model $f$, loss function $\ell$, maximum perturbation budget $\epsilon_{max}$, minimum adversarial loss $\rho$, perturbation budget $\epsilon$, perturbation budget step size $\tau$, SA-PGD step $K$, SA-PGD step size $\alpha$
2: **Output:** $x'$
3: $x' \leftarrow x; \epsilon \leftarrow 0$
4: **if** $\ell(f(x'), y) \geq \rho$ **then**
5:     **break**
6: **else**
7:     **while** $\epsilon < \epsilon_{max}$ **do**
8:         $\epsilon \leftarrow \epsilon + \tau$
9:         **for** $k = 1, ..., K$ **do**
10:           $x' \leftarrow \Pi_\epsilon(\alpha \cdot \text{sign}(\nabla_{x'}\ell(f(x'), y)) + x')$
11:           **if** $\ell(f(x'), y) \geq \rho$ **then**
12:             **return** $x'$
13:           **end if**
14:         **end for**
15:     **end while**
16: **end if**

---

**Algorithm 2** Strength-Adaptive Adversarial Training (SAAT)

---

1: **Input:** network $f_\theta$, training dataset $\mathcal{S} = \{(x_i, y_i)\}_{i=1}^n$, learning rate $\eta$, number of epochs $T$, batch size $m$, number of batches $M$
2: **Output:** adversarially robust network $f_\theta$ with a target degree of robustness disparity
3: **for** epoch $= 1, ..., T$ **do**
4:     **for** mini-batch $= 1, ..., M$ **do**
5:         Sample a mini-batch $\{(x_i, y_i)\}_{i=1}^m$ from $\mathcal{S}$
6:         **for** $i = 1, ..., m$ (in parallel) **do**
7:           Obtain adversarial data $x'_i$ of $x_i$ by Algorithm 1
8:         **end for**
9:         $\theta \leftarrow \theta - \eta\frac{1}{m}\sum_{i=1}^m \nabla_\theta\ell(f_\theta(x'_i), y_i)$
10:     **end for**
11: **end for**

---

computational cost and slows the speed, even though it ensures that the optimal perturbation budget will be estimated.

A maximally allowed perturbation budget ($\epsilon_{max}$) is introduced: we observe that even with the largest perturbation (e.g., $\epsilon = 255/255$), there are still some examples (outliers) that fail to satisfy the minimum adversarial loss constraint. Considering that the pixel values are strictly sampled in $[0, 255/255]$, it is necessary to introduce a maximum perturbation budget to avoid infinite loops.

Algorithm 1 is our strength-adaptive PGD method (SA-PGD), which returns the generated adversarial training data. Algorithm 2 is the proposed strength-adaptive adversarial training (SAAT). SAAT leverages Algorithm 1 for obtaining the qualified adversarial data to optimize the model parameters.

## 4 EXPERIMENTS

In this section, we conduct comprehensive experiments to evaluate the effectiveness of SAAT including its experimental setup (in Section 4.1), algorithm analysis (in Section 4.2), robustness evaluation (in Section 4.3), and the performance under different model capacities (in Section 4.4).

### 4.1 EXPERIMENTAL SETUP

Our code is implemented on the open source PyTorch framework with a single NVIDIA A100-SXM4-40GB GPU. The code as well as related models will be released for public use and verification. We conduct experiments on $\ell_\infty$ threat model and follow the hyper-parameter setting of Rice et al. (2020) for a fair comparison with the state-of-the-art AT methods. For training, the network is trained for 200 epochs using SGD with momentum 0.9, weight decay $5 \times 10^{-4}$, and an initial learning rate of 0.1. The learning rate is divided by 10 at the 100-th and 150-th epoch, re-

spectively. Conventional data augmentation including random crops with 4 pixels of padding and random horizontal flips are applied. For adversary, we use SA-PGD (Algorithm 1) to generate adversarial training data. The step size $\alpha = 2/255$ is used following standard PGD (Madry et al., 2017). We adopt the perturbation budget step size $\tau$ consistent with the SA-PGD step size $\alpha$, such as $\tau = 2/255$, and SA-PGD step $K = 3$, which is to make the adversarial data sufficiently attacked under the updated perturbation budget. For robustness evaluation, the output model is tested under a series of adversaries, including natural data, PGD (Madry et al., 2017), and Auto Attack (Croce & Hein, 2020b). Among them, natural accuracy and PGD accuracy can intuitively reflect the network's robustness disparity. And AA is an ensemble of complementary attacks, consisting of three white-box attacks (APGD-CE (Croce & Hein, 2020b), APGD-DLR (Croce & Hein, 2020b), and FAB (Croce & Hein, 2020a)) and a black-box attack (Square Attack (Andriushchenko et al., 2020)). AA regards networks to be robust only if the model correctly classify adversarial data with all types of attacks, which is among the most reliable evaluation of adversarial robustness to date.

## 4.2 ANALYSIS OF THE PROPOSED ALGORITHM

We delve into SAAT to investigate its each component, including the role of minimum adversarial loss $\rho$ and the impact of maximum perturbation budget $\epsilon_{max}$. All analysis experiments are conducted using PreAct ResNet-18 (He et al., 2016) on CIFAR10 dataset (Krizhevsky et al., 2009).

**The role of minimum adversarial loss $\rho$.** We empirically investigate the role of minimum adversarial loss by using different $\rho$ to generate adversarial training data, where $\epsilon_{max}$ is assigned a sufficiently large value, such as $\epsilon_{max} = 128/255$. The minimum adversarial loss $\rho$ for SAAT varies from 0 to 2.2, and the evaluation results are summarized in Figure 3 (a). It is observed that the model's degree of robustness disparity is well-correlated with the adversarial training loss $\rho$. When $\rho = 0$, there is a large performance gap between the natural accuracy and robust accuracy. This performance gap keeps decreasing as $\rho$ increases. And when $\rho = 2.2$, the robust accuracy is almost the same as the natural accuracy. Note that SAAT fails to converge when $\rho > 2.2$, which might be explained by the fact that robust accuracy should be lower than natural accuracy for any adversarially-trained networks. The clear correlation between the minimum training loss and degree of robustness disparity enables our SAAT to flexibly control the performance gap in output networks. Moreover, we observed that $\rho$ is also closely related to the robust generalization gap. The learning cruves of SAAT with different $\rho$ is shown in Figure 3 (c), and their robust generalization gap is summarized in Figure 3 (d). It can be seen that the robust accuracy is always in sync with the natural accuracy during the training phase and there is no significant robustness degradation as in Figure 1(c). When $\rho = 1.5$, the robust generalization gap is already very small.

**The impact of maximum perturbation budget $\epsilon_{max}$.** We further investigate the impact of introduced maximum perturbation budget, by comparing the robustness performance of models trained using different $\epsilon_{max}$. Given a fixed $\rho$, such as $\rho = 1.5$, the value of $\epsilon_{max}$ varies from 0 to 128/255, and the evaluation results are summarized in Figure 3 (b). It can be observed that when $\epsilon_{max}$ is small, increasing $\epsilon_{max}$ leads to a flatter degree of robustness disparity, which indicates that more model capacity is allocated to defend against adversarial attacks. As expected, when $\epsilon_{max}$ is greater than a certain value, the degree of robustness disparity begins to maintain a plateau, which infers that most of the adversarial training data already met the minimum adversarial loss constraint, so the network tends to maintain steady robustness disparity bound by $\rho$. $\epsilon_{max}$ adjusts the robustness disparity within the plateau constrained by $\rho$, which reflects a tradeoff between the natural accuracy and adversarial robustness and suggests that our $\epsilon_{max}$ helps adjust this tradeoff. Notably, the observed tradeoff is not inherent in adversarial training but a consequence of model capacity scheduling. Note that the size of $\epsilon_{max}$ is also influenced by minimum adversarial loss $\rho$. In the following section, we fine-tune both $\epsilon_{max}$ and $\rho$ for the robustness evaluation of SAAT.

## 4.3 ROBUSTNESS EVALUATION

Compared with the standard AT, the difference of SAAT mainly lies in weakening the easy-to-attack samples (the adversarial loss is higher than the minimal adversarial loss) and enhancing the hard-to-attack samples (the adversarial loss is lower than the minimal adversarial loss). In this part, we investigate their respective effects on network adversarial robustness on two classic baselines: AT and AWP, where AWP can suppress robust overfitting and achieve state-of-the-art adversarial robust-

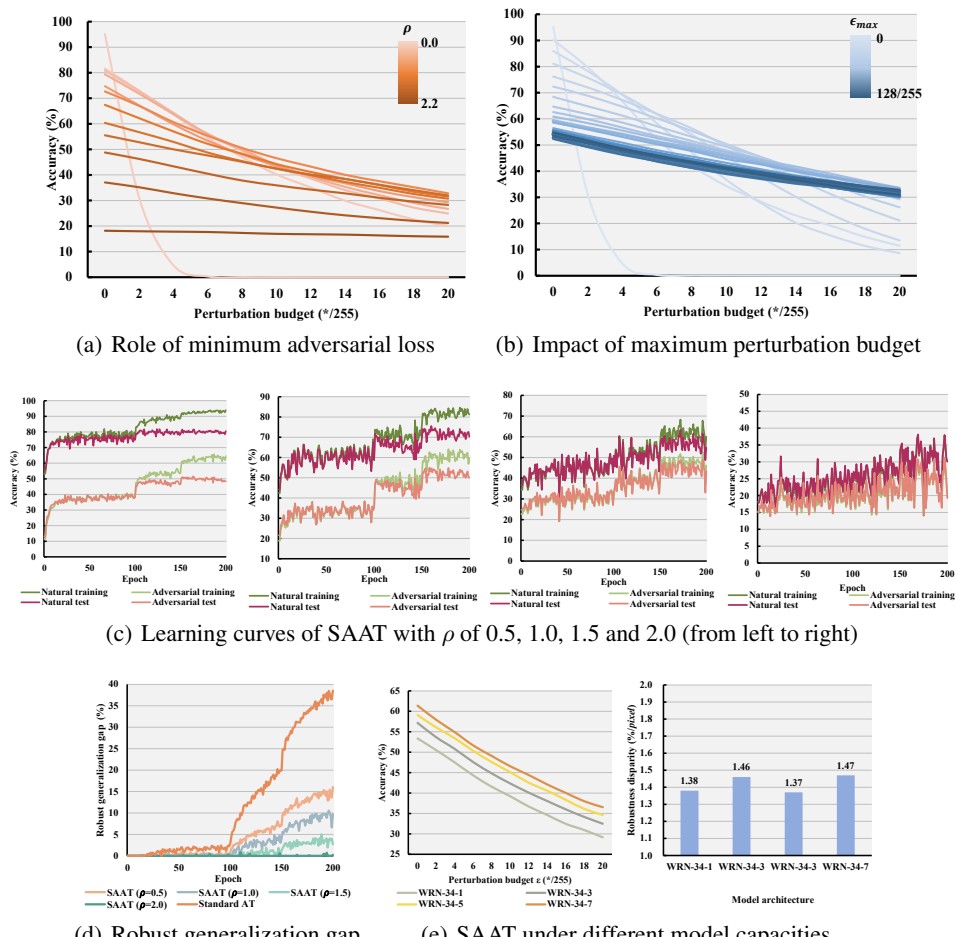

Figure 3: Robustness evaluation of SAAT with (a) varied $\rho$ and (b) varied $\epsilon_{max}$; The (c) learning curve and (d) robust generalization gap with different $\rho$; (e) robustness disparity on networks with different model capacities.

ness. Specifically, we process the easy-to-attack samples and the hard-to-attack samples separately, denoted as $\text{SAAT}_{down}$ and $\text{SAAT}_{up}$, respectively. For $\text{SAAT}_{down}$, the maximum adversarial budget remains the same as the standard AT, e.g., $\epsilon_{max} = 8$, and the minimal adversarial loss is set to 1.5. The evaluation results are shown in Table 1. As expected, weakening the easy-to-attack samples will make adversarial training inclined to the weak attack defense, which significantly increase the natural accuracy and can maintain the PGD accuracy, but it essentially not only aggravates the robustness disparity, but also degrades the model adversarial robustness (in term of AA). For $\text{SAAT}_{up}$, by increasing the maximum adversarial budget and minimal adversarial loss, more model capacity is used to defend against adversarial attacks, which significantly alleviates the robustness disparity of the output network and further enhances the model adversarial robustness. The performance evaluation of $\text{SAAT}_{up}$ across different datasets, different model structures and different AT methods is provided in Appendix A, where $\text{SAAT}_{up}$ boosts adversarial robustness under all settings, demonstrating the effectiveness of the proposed method. Note that the robust accuracy of $\text{SAAT}_{up}$ at $\epsilon = 8$ first increases and then decreases. This is defensible because as $\epsilon$ and $\rho$ increase, the robustness disparity of the output model is getting flatter, and the adversarial robustness will also be migrated to a relatively larger perturbation budget. In addition, it can be observed that the performance gap between the last checkpoint and the best checkpoint of $\text{SAAT}_{up}$ is shrinking, which illustrates that enhancing the hard-to-attack samples can also effectively alleviate robust overfitting.

## 4.4 PERFORMANCE UNDER DIFFERENT MODEL CAPACITIES

We extend SAAT to networks with different model capacities. Specifically, we perform SAAT on a series of Wide ResNet-34-x networks, where x is 1, 3, 5, and 7 respectively. Note that the larger the x, the larger the model capacity. $\rho$ is fixed at 1.5 and $\epsilon_{max}$ is sufficiently large, such

Table 1: Robustness evaluation of $SAAT_{down}$ and $SAAT_{up}$ under adversarial budget $\epsilon = 8$. We omit the standard deviations of 3 runs as they are very small (the Natural is $< 0.8\%$, the PGD-20 is $< 0.5\%$, and the AA is $< 0.15\%$).

| Method | Best | | | Last | | |
|---|---|---|---|---|---|---|
| | Natural | PGD-20 | AA | Natural | PGD-20 | AA |
| AT | 82.02 | 52.59 | 48.23 | 83.89 | 45.28 | 42.88 |
| $SAAT_{down}$ ($\epsilon_{max} = 8, \rho = 1.5$) | 82.97 | 52.87 | 45.54 | 85.31 | 47.06 | 42.77 |
| $SAAT_{up}$ ($\epsilon_{max} = 9, \rho = 1.5$) | 81.04 | 53.00 | 47.72 | 82.82 | 46.49 | 43.66 |
| $SAAT_{up}$ ($\epsilon_{max} = 10, \rho = 1.5$) | 80.81 | 53.63 | 47.83 | 82.14 | 48.14 | 44.67 |
| $SAAT_{up}$ ($\epsilon_{max} = 11, \rho = 1.5$) | 77.99 | 54.38 | 47.90 | 80.55 | 48.86 | 44.85 |
| $SAAT_{up}$ ($\epsilon_{max} = 12, \rho = 1.5$) | 77.50 | 54.78 | 48.04 | 79.44 | 50.03 | 45.42 |
| $SAAT_{up}$ ($\epsilon_{max} = 13, \rho = 1.5$) | 77.40 | 55.21 | 48.26 | 78.48 | 51.68 | 45.55 |
| $SAAT_{up}$ ($\epsilon_{max} = 14, \rho = 1.5$) | 76.80 | 55.59 | **48.61** | 77.44 | 52.80 | **46.46** |
| $SAAT_{up}$ ($\epsilon_{max} = 15, \rho = 1.5$) | 76.62 | 56.18 | 48.45 | 76.36 | 53.62 | 46.12 |
| $SAAT_{up}$ ($\epsilon_{max} = 16, \rho = 1.5$) | 75.54 | 56.22 | 48.12 | 75.52 | 54.17 | 46.07 |
| $SAAT_{up}$ ($\epsilon_{max} = 17, \rho = 1.5$) | 74.21 | 56.35 | 47.95 | 74.49 | 54.33 | 45.73 |
| $SAAT_{up}$ ($\epsilon_{max} = 18, \rho = 1.5$) | 73.89 | **56.49** | 47.64 | 73.55 | **54.79** | 45.66 |
| $SAAT_{up}$ ($\epsilon_{max} = 19, \rho = 1.5$) | 73.68 | 56.12 | 46.63 | 72.67 | 54.72 | 45.46 |
| $SAAT_{up}$ ($\epsilon_{max} = 20, \rho = 1.5$) | 71.89 | 56.11 | 45.96 | 72.15 | 54.62 | 44.94 |
| $SAAT_{up}$ ($\epsilon_{max} = 14, \rho = 1.3$) | 78.40 | 54.87 | 48.08 | 78.49 | 51.68 | 45.77 |
| $SAAT_{up}$ ($\epsilon_{max} = 14, \rho = 1.4$) | 77.45 | 55.04 | 48.23 | 78.20 | 52.40 | 45.79 |
| $SAAT_{up}$ ($\epsilon_{max} = 14, \rho = 1.5$) | 76.80 | 55.59 | 48.61 | 77.44 | 52.80 | 46.46 |
| $SAAT_{up}$ ($\epsilon_{max} = 14, \rho = 1.6$) | 76.48 | 56.08 | 48.76 | 77.16 | 53.11 | 46.57 |
| $SAAT_{up}$ ($\epsilon_{max} = 14, \rho = 1.7$) | 76.37 | **56.31** | **48.86** | 76.38 | **53.86** | **47.17** |
| $SAAT_{up}$ ($\epsilon_{max} = 14, \rho = 1.8$) | 75.51 | 56.18 | 48.85 | 76.11 | 53.42 | 46.73 |
| $SAAT_{up}$ ($\epsilon_{max} = 14, \rho = 1.9$) | 75.10 | 55.58 | 48.59 | 75.46 | 53.20 | 46.53 |
| $SAAT_{up}$ ($\epsilon_{max} = 14, \rho = 2.0$) | 74.63 | 55.62 | 48.36 | 75.06 | 53.12 | 46.29 |
| AWP-AT | 81.47 | 55.54 | 49.96 | 80.20 | 54.88 | 49.28 |
| AWP-$SAAT_{down}$ ($\epsilon_{max} = 8, \rho = 1.5$) | 83.74 | 56.97 | 48.38 | 82.85 | 55.16 | 46.99 |
| AWP-$SAAT_{up}$ ($\epsilon_{max} = 9, \rho = 1.5$) | 79.55 | 55.95 | **50.32** | 78.08 | 54.93 | **49.36** |
| AWP-$SAAT_{up}$ ($\epsilon_{max} = 10, \rho = 1.5$) | 78.14 | 56.20 | 49.98 | 76.66 | 55.12 | 48.89 |
| AWP-$SAAT_{up}$ ($\epsilon_{max} = 11, \rho = 1.5$) | 77.17 | 56.43 | 49.87 | 74.98 | 55.26 | 48.74 |
| AWP-$SAAT_{up}$ ($\epsilon_{max} = 12, \rho = 1.5$) | 76.00 | **56.76** | 49.43 | 73.49 | **55.33** | 48.22 |
| AWP-$SAAT_{up}$ ($\epsilon_{max} = 9, \rho = 1.6$) | 79.52 | 56.00 | 50.35 | 78.06 | 54.97 | 49.12 |
| AWP-$SAAT_{up}$ ($\epsilon_{max} = 9, \rho = 1.7$) | 79.49 | **56.25** | **50.67** | 77.91 | **55.22** | **49.29** |
| AWP-$SAAT_{up}$ ($\epsilon_{max} = 9, \rho = 1.8$) | 79.32 | 56.05 | 50.26 | 77.88 | 54.98 | 49.18 |
| AWP-$SAAT_{up}$ ($\epsilon_{max} = 9, \rho = 1.9$) | 78.96 | 55.97 | 50.20 | 77.81 | 54.69 | 48.92 |

as $\epsilon_{max} = 128/255$. The evaluation results are summarized in Figure 3 (e). It can be observed that the output network can maintain a steady degree of robustness disparity across different model capacities. Such observations exactly reflect the nature of our approach that SAAT is able to adapt to networks of various model capacities.

## 5 CONCLUSION

We present a strength-adaptive adversarial training (SSAT) method in this paper. The proposed approach distinguish itself from others by using the minimum adversarial loss constraint for generating adversarial training data, which is adaptive to the dynamic training schedule and networks of various model capacities. We show that adversarial training loss is well-correlated with the degree of robustness disparity and robust generalization gap, empirically verify that SAAT can effectively alleviate robustness overfitting, mitigate the robustness disparity of output networks and further enhance the model adversarial robustness by adjusting the tradeoff of adversarial training, demonstrating the effectiveness of the proposed approach. We hope that the robustness disparity we offer can further improve the completeness of adversarial robustness evaluation methods (Carlini et al., 2019) and expect more new techniques to be proposed to achieve learning a fully robust network.

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

## A  APPENDIX

In this part, we conduct extended comparative experiments. Specifically, we conduct experiments under different datasets, different model structures and different AT methods. The results are summarized in Table 2 and Table 3. Experimental results show that the proposed method can achieve higher adversarial robustness under all settings, demonstrating the effectiveness of the proposed approach.

Table 2: Robustness evaluation of $\text{SAAT}_{\text{up}}$ (best checkpoint) under adversarial budget $\epsilon = 8$ across different datasets, different network structures and different AT methods, where PR-18 indicates PreAct ResNet-18 and WRN-34-10 indicates Wide ResNet-34-10.

| Dataset | Network | Basic AT | Method | Best | | |
| --- | --- | --- | --- | --- | --- | --- |
| | | | | Natural | PGD-20 | AA |
| CIFAR10 | PR-18 | AT | AT | 82.02 | 52.59 | 48.23 |
| CIFAR10 | PR-18 | AT | $\text{SAAT}_{\text{up}}$ | 76.37 | **56.31** | **48.86** |
| CIFAR10 | PR-18 | AT | AWP-AT | 81.47 | 55.54 | 49.96 |
| CIFAR10 | PR-18 | AT | $\text{AWP-SAAT}_{\text{up}}$ | 79.49 | **56.25** | **50.67** |
| CIFAR100 | PR-18 | AT | AT | 55.95 | 28.84 | 24.71 |
| CIFAR100 | PR-18 | AT | $\text{SAAT}_{\text{up}}$ | 53.20 | **29.29** | **25.25** |
| CIFAR100 | PR-18 | AT | AWP-AT | 56.64 | 32.02 | 26.43 |
| CIFAR100 | PR-18 | AT | $\text{AWP-SAAT}_{\text{up}}$ | 54.36 | **32.09** | **26.98** |
| CIFAR10 | PR-18 | TRADES | TRADES | 81.09 | 52.70 | 48.91 |
| CIFAR10 | PR-18 | TRADES | $\text{SATRADES}_{\text{up}}$ | 81.76 | **53.44** | **50.04** |
| CIFAR10 | PR-18 | TRADES | AWP-TRADES | 81.64 | 55.52 | 51.38 |
| CIFAR10 | PR-18 | TRADES | $\text{AWP-SATRADES}_{\text{up}}$ | 80.76 | **55.98** | **51.72** |
| CIFAR10 | WRN-34-10 | AT | AT | 85.47 | 54.87 | 51.70 |
| CIFAR10 | WRN-34-10 | AT | $\text{SAAT}_{\text{up}}$ | 79.94 | **58.52** | **52.36** |
| CIFAR10 | WRN-34-10 | AT | AWP-AT | 86.00 | 58.78 | 54.06 |
| CIFAR10 | WRN-34-10 | AT | $\text{AWP-SAAT}_{\text{up}}$ | 83.55 | **59.20** | **54.64** |

Table 3: Robustness evaluation of $\text{SAAT}_{\text{up}}$ (last checkpoint) under adversarial budget $\epsilon = 8$ across different datasets, different network structures and different AT methods, where PR-18 indicates PreAct ResNet-18 and WRN-34-10 indicates Wide ResNet-34-10.

| Dataset | Network | Basic AT | Method | Last | | |
| --- | --- | --- | --- | --- | --- | --- |
| | | | | Natural | PGD-20 | AA |
| CIFAR10 | PR-18 | AT | AT | 83.89 | 45.28 | 42.88 |
| CIFAR10 | PR-18 | AT | $\text{SAAT}_{\text{up}}$ | 76.38 | **53.86** | **47.17** |
| CIFAR10 | PR-18 | AT | AWP-AT | 80.20 | 54.88 | 49.28 |
| CIFAR10 | PR-18 | AT | $\text{AWP-SAAT}_{\text{up}}$ | 77.91 | **55.22** | **49.29** |
| CIFAR100 | PR-18 | AT | AT | 56.75 | 21.12 | 19.35 |
| CIFAR100 | PR-18 | AT | $\text{SAAT}_{\text{up}}$ | 54.52 | **22.32** | **20.12** |
| CIFAR100 | PR-18 | AT | AWP-AT | 57.18 | 31.64 | 26.45 |
| CIFAR100 | PR-18 | AT | $\text{AWP-SAAT}_{\text{up}}$ | 55.08 | **31.72** | **27.04** |
| CIFAR10 | PR-18 | TRADES | TRADES | 82.40 | 49.98 | 47.02 |
| CIFAR10 | PR-18 | TRADES | $\text{SATRADES}_{\text{up}}$ | 81.30 | **51.13** | **47.99** |
| CIFAR10 | PR-18 | TRADES | AWP-TRADES | 81.87 | 55.41 | 51.27 |
| CIFAR10 | PR-18 | TRADES | $\text{AWP-SATRADES}_{\text{up}}$ | 80.64 | **55.62** | **51.60** |
| CIFAR10 | WRN-34-10 | AT | AT | 86.16 | 46.94 | 45.68 |
| CIFAR10 | WRN-34-10 | AT | $\text{SAAT}_{\text{up}}$ | 80.19 | **55.47** | **49.98** |
| CIFAR10 | WRN-34-10 | AT | AWP-AT | 87.08 | 57.96 | 53.41 |
| CIFAR10 | WRN-34-10 | AT | $\text{AWP-SAAT}_{\text{up}}$ | 84.05 | **58.76** | **54.16** |

