# OpenReview forum: "Strength-Adaptive Adversarial Training"
_ICLR.cc/2023/Conference — Submitted to ICLR 2023_

### Official Review · Reviewer_AoDK · 2022-10-19

**Confidence:** 4
**Clarity, Quality, Novelty And Reproducibility:** The manuscript is well written and cl…
**Correctness:** 3
**Technical Novelty And Significance:** 2
**Empirical Novelty And Significance:** 2
**Recommendation:** 5

**Strength And Weaknesses:**

Strength:
1. Well written and easy to follow
2. Idea is straight forward

Weaknesses:
1. A little novelty. For me, setting a minimum adversarial loss during adversarial training is not exciting.

2. Weak experiments.
(a) There is just a trade-off between adversarial accuracy (under auto attack) and clean accuracy in the experiments, i.e., increase (adaptive) adversarial attack strength during training -> improve test adversarial accuracy and damage test clean accuracy.
(b) In the current development of adversarial training, Wide ResNet-34-10 on CIFAR-10/100 is necessary, where TRADES-AWP can get a better performance under auto attack. I wonder the performance of SAAT under these settings. (i.e., the results of Tab. 1 with WRN-34-10 and TRADES-AWP).

3. Experiment-based work without deep analysis.

**Summary Of The Paper:**

This work proposes adversarial training with a threshold of minimum adversarial loss, i.e., while the training adversarial example cannot reach the minimum loss, expand its attack radius.

**Summary Of The Review:**

This is an experiment based work without deep (theoretical) analysis, it only presents weak empirical results with a straight-forward idea, i.e., a trade-off between clean accuracy and adversarial accuracy (under AA) with SAAT. For this reason, I think it is below the bar of ICLR.

---

> ### Author Response · Authors · 2022-11-13
> **Response to Reviewer AoDK**
>
> Thanks for your careful and valuable comments. We will explain your concerns point by point.
>
> **Q1:**  A little novelty & experiment-based work without deep analysis.
>
> **A1:** Our method uses adversarial loss mechanism to adaptively adjust the adversarial budget, and is also the first adaptive budget-based method to achieve robustness better than Standard AT (works on adaptive adversarial budgets [1,2] both degrade adversarial robustness compared to Standard AT), which we believe can bring some new knowledge to the community. At the same time, Our paper proposes and defines the robustness disparity, which helps to promote the development of robustness in adversarial training to align with human perception. Moreover, our method provides a scheme for training networks with similar robustness to human perception under the condition of limited model capacity. These are the contributions of our paper, which hopefully alleviate concerns about the novelty of our paper.
>
> [1] Yogesh Balaji, Tom Goldstein, Judy Hoffman: Instance adaptive adversarial training: Improved accuracy tradeoffs in neural nets. CoRR abs/1910.08051 (2019)
>
> [2] Gavin Weiguang Ding, Yash Sharma, Kry Yik Chau Lui, Ruitong Huang: MMA Training: Direct Input Space Margin Maximization through Adversarial Training. ICLR 2020
>
> **Q2:**  Weak experiments.
>
> **A2:** (a) The degradation of natural accuracy is defensible, in other words, it's actually the advantage of our method. According to the research in [1], current deep learning network has high natural accuracy, mainly because the network has learned some non-robust features (or features that are not related to semantics). Our method constrains the robustness disparity of the output network, which degrades the natural accuracy, but also brings the natural accuracy and robust accuracy closer, which just reflects the nature of our training method that makes the network pay more attention to robust features (or semantic features), thereby outputting a robust classifier that aligns with human perception.
> (b) We conducted a series of experiments across different network structures, different datasets and different AT baselines. The results are summarized in the following table. Our method consistently improves adversarial robustness in all settings. We will add these experimental results to the paper, hoping that these results can alleviate concerns about our experiments.
>
> [1] Dimitris Tsipras, Shibani Santurkar, Logan Engstrom, Alexander Turner, Aleksander Madry: Robustness May Be at Odds with Accuracy. ICLR 2019
>
> |Dataset|Network|Basic AT Framework| Method|Natural (Best)| PGD-20 (Best)| AA (Best)|Natural (Last)| PGD-20 (Last)| AA (Last)|
> | :-----| :-----| :-----|:-----| ----: | ----: |----: | ----: |----: | :----: |
> |CIFAR10|PreAct ResNet 18|AT|  $\mathrm{AT}$ |82.02 | 52.59| 48.23| 83.89| 45.28| 42.88|
> |CIFAR10|PreAct ResNet 18|AT| $\mathrm{SAAT_{up}}$ |76.37 | **56.31**| **48.86**| 76.38| **53.86**| **47.17**|
> |CIFAR10|PreAct ResNet 18|AT| $\mathrm{AWP-AT}$ |81.47 | 55.54| 49.96| 80.20| 54.88| 49.28|
> |CIFAR10|PreAct ResNet 18|AT| $\mathrm{AWP-SAAT_{up}}$ |79.49 | **56.25**| **50.67**| 77.91| **55.22**| **49.29**|
> |CIFAR100|PreAct ResNet 18|AT| $\mathrm{AT}$ |55.95 | 28.84| 24.71| 56.75| 21.12| 19.35|
> |CIFAR100|PreAct ResNet 18|AT| $\mathrm{SAAT_{up}}$ |53.20 | **29.29**| **25.25**| 54.52| **22.32**| **20.12**|
> |CIFAR100|PreAct ResNet 18|AT| $\mathrm{AWP-AT}$ |56.64 | 32.02| 26.43| 57.18| 31.64| 26.45|
> |CIFAR100|PreAct ResNet 18|AT| $\mathrm{AWP-SAAT_{up}}$ |54.36 | **32.09**| **26.98**| 55.08| **31.72**| **27.04**|
> |CIFAR10|PreAct ResNet 18|TRADES| $\mathrm{TRADES}$ |81.09 | 52.70| 48.91| 82.40| 49.98| 47.02|
> |CIFAR10|PreAct ResNet 18|TRADES| $\mathrm{SATRADES_{up}}$ |81.76 | **53.44**| **50.04**| 81.30| **51.13**| **47.99**|
> |CIFAR10|PreAct ResNet 18|TRADES| $\mathrm{AWP-TRADES}$ |81.64 | 55.52| 51.38| 81.87| 55.41| 51.27|
> |CIFAR10|PreAct ResNet 18|TRADES| $\mathrm{AWP-SATRADES_{up}}$ |80.76 | **55.98**| **51.72**| 80.64| **55.62**| **51.60**|
> |CIFAR10|Wide ResNet-34-10|AT| $\mathrm{AT}$ |85.47 | 54.87| 51.70| 86.16| 46.94| 45.68|
> |CIFAR10|Wide ResNet-34-10|AT| $\mathrm{SAAT_{up}}$ |79.94 | **58.52**| **52.36**| 80.19| **55.47**| **49.98**|
> |CIFAR10|Wide ResNet-34-10|AT| $\mathrm{AWP-AT}$ |86.00 | 58.78| 54.06| 87.08| 57.96| 53.41|
> |CIFAR10|Wide ResNet-34-10|AT| $\mathrm{AWP-SAAT_{up}}$ |83.55 | **59.20**| **54.64**| 84.05| **58.76**| **54.16**|
>
> The above is our explanation. If you have anything unclear or have further comments, please feel free to give us feedback and look forward to discussing with you. Thanks again for your time.

---

> > ### Author Response · Authors · 2022-11-15
> > **Reminder - Discussion Stage 1 closing soon - 18 November**
> >
> > We much appreciate the time and effort that you have dedicated to reviewing our manuscript. Just a quick reminder that we have explained your concerns about our manuscript, and please let us know if you have any questions about these explanations or new comments. We look forward to your feedback.

---

> > ### Author Response · Authors · 2022-11-17
> > **Final Reminder - Discussion Stage 1 closing soon - 18 November**
> >
> > A final reminder that discussion stage 1 is closing soon. Please let us know if you have any further questions about our explanations, and thanks again for the time and effort that you have dedicated to reviewing our manuscript.

---

### Official Review · Reviewer_snup · 2022-10-24

**Confidence:** 4
**Correctness:** 3
**Technical Novelty And Significance:** 2
**Empirical Novelty And Significance:** Not applicable
**Recommendation:** 3

**Clarity, Quality, Novelty And Reproducibility:**

This paper is well-written and clear. The proposed method is very naive. This paper sounds technical and
can be reproduced.

**Strength And Weaknesses:**

**Strengths:**

1. The proposed idea is very straightforward and clean, although I have some questions regarding the proposed pseudo-codes.
2. Experimental results show that SAAT can achieve consistent yet marginal improvements over vanilla
adversarial training and AWP.

**Weaknesses:**
1. Adopting adaptive attacking budget during adversarial training has been highlighted by many previous
work. The proposed SAAT is just a simple early stop (loss value) based on some empirical observations,
which in my personal view is not novel.
2. In Alg.1, considering that step size $\alpha$ is the same as perturbation budget step size $\tau(2/255)$, what is
the meaning of enlarging epsilon by $\tau$ every attacking step? The real adversarial perturbation will be
bounded by the step size (if ignoring [min, max] clipping). So in my view, the adaptive attack budget is
just the normal PGD using $\epsilon = \epsilon_{max}$. Besides, the break in Line 12 only jumps up of the inner for-loop.
Does this mean that attacking will continue after enlarging the epsilon for this adv. example whose loss
has already been greater than $\rho$?
3. By setting SA-PGD step K = 3, the proposed SAAT might adopt much more attack steps to train
their models. For example, with $\epsilon_{max}$ = 8/255, $\tau$ = 2/255, the actual attack steps might be at most
$\epsilon/T*K = 12$. When $\epsilon_{max}$ = 14/255 (common setting in this paper’s experiments), the attack steps
might be at most 21. A lot of computational costs will be introduced in SAAT.
4. The proposed SAAT doesn’t show any improvements under common adversarial evaluation settings
(eps=8/255) and only marginal improvements can be observed on vanilla adversarial training and AWP
under very specific settings. Besides, SAAT hurts clean accuracy very much. Only experiments on
CIFAR-10 and Resnet-18 are provided. The authors should conduct experiments on other architectures
with larger capacities and benchmarks (CIFAR-100).

**Summary Of The Paper:**

This paper proposes Strength-Adaptive Adversarial Training (SAAT) to improve adversarial training. Specifically, SAAT adopts a dynamic and adaptive strategy to control the adversarial attack budget. Instead of setting a fixed adversarial budget, this paper proposes to enlarge the budget while attacking. The proposed SAAT is evaluated on CIFAR-10 with Resnet-18. Some marginal improvements are observed.

**Summary Of The Review:**

The proposed method is straightforward, clean, and naive. Performances are my biggest concern. SAAT
doesn’t show any improvements under common adversarial evaluation settings but hurts a lot of baseline
performances.

---

> ### Author Response · Authors · 2022-11-13
> **Response to Reviewer snup (Part 2)**
>
> **Q4:** About the experiments.
>
> **A4:** Yes, SAAT does degrade natural accuracy. But it's defensible, in other words, it's actually our advantage. According to the research in [1], current deep learning network has high natural accuracy, mainly because the network has learned some non-robust features (or features that are not related to semantics). Our method can make the natural accuracy and robust accuracy closer, which just reflects the nature of our method that makes the network pay more attention to robust features (or semantic features), so as to output classifier that approximate human perception. Then regarding the experimental performance, we conducted a series of experiments across different network structures, different datasets and different AT baselines. The results are summarized in the following table. Our method consistently improves adversarial robustness in all settings. We will add these experimental results to the paper, hoping that these results can alleviate concerns about the experimental performance of our method.
>
> [1] Dimitris Tsipras, Shibani Santurkar, Logan Engstrom, Alexander Turner, Aleksander Madry: Robustness May Be at Odds with Accuracy. ICLR 2019
>
> |Dataset|Network|Basic AT Framework| Method|Natural (Best)| PGD-20 (Best)| AA (Best)|Natural (Last)| PGD-20 (Last)| AA (Last)|
> | :-----| :-----| :-----|:-----| ----: | ----: |----: | ----: |----: | :----: |
> |CIFAR10|PreAct ResNet 18|AT|  $\mathrm{AT}$ |82.02 | 52.59| 48.23| 83.89| 45.28| 42.88|
> |CIFAR10|PreAct ResNet 18|AT| $\mathrm{SAAT_{up}}$ |76.37 | **56.31**| **48.86**| 76.38| **53.86**| **47.17**|
> |CIFAR10|PreAct ResNet 18|AT| $\mathrm{AWP-AT}$ |81.47 | 55.54| 49.96| 80.20| 54.88| 49.28|
> |CIFAR10|PreAct ResNet 18|AT| $\mathrm{AWP-SAAT_{up}}$ |79.49 | **56.25**| **50.67**| 77.91| **55.22**| **49.29**|
> |CIFAR100|PreAct ResNet 18|AT| $\mathrm{AT}$ |55.95 | 28.84| 24.71| 56.75| 21.12| 19.35|
> |CIFAR100|PreAct ResNet 18|AT| $\mathrm{SAAT_{up}}$ |53.20 | **29.29**| **25.25**| 54.52| **22.32**| **20.12**|
> |CIFAR100|PreAct ResNet 18|AT| $\mathrm{AWP-AT}$ |56.64 | 32.02| 26.43| 57.18| 31.64| 26.45|
> |CIFAR100|PreAct ResNet 18|AT| $\mathrm{AWP-SAAT_{up}}$ |54.36 | **32.09**| **26.98**| 55.08| **31.72**| **27.04**|
> |CIFAR10|PreAct ResNet 18|TRADES| $\mathrm{TRADES}$ |81.09 | 52.70| 48.91| 82.40| 49.98| 47.02|
> |CIFAR10|PreAct ResNet 18|TRADES| $\mathrm{SATRADES_{up}}$ |81.76 | **53.44**| **50.04**| 81.30| **51.13**| **47.99**|
> |CIFAR10|PreAct ResNet 18|TRADES| $\mathrm{AWP-TRADES}$ |81.64 | 55.52| 51.38| 81.87| 55.41| 51.27|
> |CIFAR10|PreAct ResNet 18|TRADES| $\mathrm{AWP-SATRADES_{up}}$ |80.76 | **55.98**| **51.72**| 80.64| **55.62**| **51.60**|
> |CIFAR10|Wide ResNet-34-10|AT| $\mathrm{AT}$ |85.47 | 54.87| 51.70| 86.16| 46.94| 45.68|
> |CIFAR10|Wide ResNet-34-10|AT| $\mathrm{SAAT_{up}}$ |79.94 | **58.52**| **52.36**| 80.19| **55.47**| **49.98**|
> |CIFAR10|Wide ResNet-34-10|AT| $\mathrm{AWP-AT}$ |86.00 | 58.78| 54.06| 87.08| 57.96| 53.41|
> |CIFAR10|Wide ResNet-34-10|AT| $\mathrm{AWP-SAAT_{up}}$ |83.55 | **59.20**| **54.64**| 84.05| **58.76**| **54.16**|
>
> The above is our explanation. If you have anything unclear or have further comments, please feel free to give us feedback and look forward to discussing with you. Thanks again for your time.

---

> > ### Author Response · Authors · 2022-11-15
> > **Reminder - Discussion Stage 1 closing soon - 18 November**
> >
> > We much appreciate the time and effort that you have dedicated to reviewing our manuscript. Just a quick reminder that we have explained your concerns about our manuscript, and please let us know if you have any questions about these explanations or new comments. We look forward to your feedback.

---

> > ### Author Response · Authors · 2022-11-17
> > **Final Reminder - Discussion Stage 1 closing soon - 18 November**
> >
> > A final reminder that discussion stage 1 is closing soon. Please let us know if you have any further questions about our explanations, and thanks again for the time and effort that you have dedicated to reviewing our manuscript.

---

> ### Author Response · Authors · 2022-11-13
> **Response to Reviewer snup (Part 1)**
>
> Thanks for your careful and valuable comments. We will explain your concerns point by point.
>
> **Q1:** Adaptive adversarial budget is not novel.
>
> **A1:** It is true that previous works have investigated adaptive adversarial budget, but we have yet to see work on adaptive adversarial budgets that can outperform standard AT in adversarial robustness (under the commonly used adversarial budget $\epsilon = 8/255$). IAAT[1] and MMA[2] are two adaptive adversarial budget methods, where IAAT improves natural accuracy but degrades adversarial robustness. MMA also degrades adversarial robustness (compared to Standard AT, see #14 of Table 1 in [3]). Our work proposes a new adaptive adversarial budget mechanism and consistently improves adversarial robustness, which we believe may bring some new knowledge to the community. Regarding the proposed method, it is true that our method is simple and straightforward, but the robustness disparity is proposed and defined in our work, and most importantly, our simple method can effectively deal with the problem of robustness disparity. Hypothetically, if we need a network with a certain robustness disparity, is there a better solution from the existing work?
>
> [1] Yogesh Balaji, Tom Goldstein, Judy Hoffman: Instance adaptive adversarial training: Improved accuracy tradeoffs in neural nets. CoRR abs/1910.08051 (2019)
>
> [2] Gavin Weiguang Ding, Yash Sharma, Kry Yik Chau Lui, Ruitong Huang: MMA Training: Direct Input Space Margin Maximization through Adversarial Training. ICLR 2020
>
> [3] Francesco Croce, Matthias Hein: Reliable evaluation of adversarial robustness with an ensemble of diverse parameter-free attacks. ICML 2020
>
> **Q2:** Questions regarding the pseudo-codes.
>
> **A2:** Yes you are right, attacking should not be continued after enlarging the epsilon for this adversarial example whose loss has already been greater than $\rho$. Line 12 of Algorithm 1 should be **return x'**. We neglected that the Algorithm 1 in the paper is designed for a single example. In the code, we also use **break**, which is equivalent to **return x'**. Because the code executes one batch size of examples at a time and we use the index to mark which examples need further attacking, like this:
>
> ```
>     loss = F.cross_entropy(output, y, reduction='none')
>     index = torch.where(loss < \rho)[0]
>     if len(index) == 0:
>         break
>     for _ in range(k):
>         ...
>         x = X[index,:,:,:]
>         d = delta[index, :, :, :]
>         ...
>         delta.data[index,:,:,:] = d
>
>         loss = F.cross_entropy(output, y, reduction='none')
>         index = torch.where(loss < \rho)[0]
>         if len(index) == 0:
>             break
> ```
>
> After replacing line 12 of Algorithm 1 with **return x'**, there should be no doubt about the meaning of increasing $\epsilon$ by $\tau$ every attack loop. We'll correct this typo, thank you for pointing it out, and sorry for the confusion.
>
> **Q3:** About the computational costs.
>
> **A3:** Yes, our SAAT will introduce more computational cost, which we have also mentioned in Section 3.3 of the paper. But we would like to emphasize that, on the one hand, it is a common belief that training a robust classifier closer to human perception requires more computational cost[1]. On the other hand, more computational cost is not the main factor of our method to improve the robustness. For a fair comparison, we run Standard AT with 21 steps and the results are shown in the following table. At the same computational cost, SAAT outperforms Standard AT. Even if the Standard AT is given more computational cost, it cannot improve the adversarial robustness of the output network, hoping that this experiments can alleviate the concerns about the computational cost.
>
> [1] Preetum Nakkiran: Adversarial Robustness May Be at Odds With Simplicity. CoRR abs/1901.00532 (2019)
>
> |Method|Iterations|Natural (Best)| PGD-20 (Best)| AA (Best)|Natural (Last)| PGD-20 (Last)| AA (Last)|
> | :-----| :---- | ----: | ----: | ----: | ----: | ----: | :----: |
> |$\mathrm{AT}$|10|82.02|52.59|48.23|83.89|45.28|42.88|
> |$\mathrm{AT}$|21|81.76|52.23|48.21|84.11|44.28|42.10|
> |$\mathrm{SAAT_{up}}$|21|76.37|**56.31**|**48.86**|76.38|**53.86**|**47.17**|

---

### Official Review · Reviewer_e7uD · 2022-10-26

**Confidence:** 4
**Clarity, Quality, Novelty And Reproducibility:** 1. Please increase the size of your i…
**Correctness:** 2
**Technical Novelty And Significance:** 2
**Empirical Novelty And Significance:** 2
**Recommendation:** 3

**Strength And Weaknesses:**

Strength
1. The algorithm is fit for networks of various model capacities.
2. Figure 2 (d) shows robustness differences between the "best" and "last" checkpoints, which is interesting.

Weakness
1. About motivation: I question the novelty of your motivation that using a dynamic perturbation budget can alleviate robustness over-fitting. In fact, Cai et al. [1] adjust the iteration numbers of PGD to control the adversarial strength. Similarly, Kim et al. [2] directly manipulate the step size of FGSM to overcome the question. At least, you should cite these works in the introduction part of your paper.

2. About methods: Your work introduces a new parameter $\rho$ and uses a heuristic method to adjust it. However, why not apply the same heuristic method to $\epsilon$ directly? So you can adjust the adversarial budget during training. I hope you can do some additional experiments to demonstrate the advantage of introducing $\rho$.

3. About methods: You claim that your method can cope with different model capacities. However, I do not see such an experiment in your paper. Could you train some models of different architectures adversarially with the same $\rho$?

4. About Experiments: Your paper bears a lack of baselines. Please add some evaluation on related works (like [1,2]) to Table 1. Note that you should evaluate these methods under the same budget.






**Summary Of The Paper:**

The paper works on adversarial training, a well-known defense for adversarial samples. It aims to address the problem of robustness over-fitting, which refers to the phenomenon that training with a fixed budget degenerates model performance. The authors propose a method called Strength-Adaptive Adversarial Training, which can update the strength of the attacks during the training process.


**Summary Of The Review:**

Although the paper has some interesting findings, it bears some severe weaknesses, like a lack of novelty, ignoring related work, and few baselines.

---

> ### Author Response · Authors · 2022-11-13
> **Response to Reviewer e7uD (Part 2)**
>
> **Q4:** About adversarial example being recognized by human.
>
> **A4:** The constraint of not being recognized by humans is for the attacker. Adversarial training, as a defense method, aims to output a classifier that is approximately as robust as the human eye. In general, for the human eye, increasing the perturbation budget will lead to a decrease in the recognition accuracy, and the degree of decrease is the concept of robustness disparity proposed in this paper. In fact, we believe that an adaptive training budget (or a perturbation budget beyond the human recognition constraints) is necessary (helpful) for the robustness of the output network to align with human perception. After all, it is pessimistic to train a classifier with robustness similar to human perception only with adversarial perturbations that humans cannot recognize, which is also one of the motivations of our approach.
>
> The above is our explanation. If you have anything unclear or have further comments, please feel free to give us feedback and look forward to discussing with you. Thanks again for your time.

---

> > ### Author Response · Authors · 2022-11-15
> > **Reminder - Discussion Stage 1 closing soon - 18 November**
> >
> > We much appreciate the time and effort that you have dedicated to reviewing our manuscript. Just a quick reminder that we have explained your concerns about our manuscript, and please let us know if you have any questions about these explanations or new comments. We look forward to your feedback.

---

> > ### Author Response · Authors · 2022-11-17
> > **Final Reminder - Discussion Stage 1 closing soon - 18 November**
> >
> > A final reminder that discussion stage 1 is closing soon. Please let us know if you have any further questions about our explanations, and thanks again for the time and effort that you have dedicated to reviewing our manuscript.

---

> ### Author Response · Authors · 2022-11-13
> **Response to Reviewer e7uD (Part 1)**
>
> Thanks for your careful and valuable comments. We will explain your concerns point by point.
>
> **Q1:** About motivation.
>
> **A1:** [1] studies catastrophic forgetting, and [2] studies catastrophic overfitting, which are essentially different from robust overfitting. Moreover, controlling the attack strength by iteration number or step size is useless for robust overfitting phenomenon. As for the iteration number, when the iteration number is insufficient, increasing the iteration number will increase the attack strength; but when the iteration number is sufficient, the effect of increasing the iteration number on the attack strength is almost negligible. For better explanation, cite a sentence from the first paragraph of Section 3 in [3]: "the loss plateaus after a few iterations, except for extremely small step sizes, which however do not translate into better results. As a consequence, judging the strength of an attack by the number of iterations is misleading".
> Then for the step size, in multi-step AT, adjusting the step size can enhance the attack strength to some extent. But this enhancement of attack strength is limited and insufficient to alleviate robust overfitting.
>
> [1] Qi-Zhi Cai, Chang Liu, Dawn Song: Curriculum Adversarial Training. IJCAI 2018
>
> [2] Hoki Kim, Woojin Lee, Jaewook Lee: Understanding Catastrophic Overfitting in Single-step Adversarial Training. AAAI 2021
>
> [3] Francesco Croce, Matthias Hein: Reliable evaluation of adversarial robustness with an ensemble of diverse parameter-free attacks. ICML 2020
>
> **Q2:** About methods.
>
> **A2:** In our algorithm, it is indeed $\epsilon$ that is adaptively adjusted, and $\rho$ is a hyperparameter that is fixed. Experiments about the advantages of introducing $\rho$ were provided in Figure 3(a) & Figure 3(c) and discussed in Section 4.2. Experiments with the same $\rho$ under different model capacities were provided in Figure 3(e) and discussed in Section 4.4.
>
> **Q3:** About experiments.
>
> **A3:** We conduct a series of experiments across different network structures, different datasets and different AT baselines. The results are summarized in the following table, and the robust accuracy (PGD-20 and AA) are evaluated under the standard adversarial budget ($\epsilon = 8/255$). Our method improves adversarial robustness in all settings. We will add these experimental results to the paper, hoping that these results can alleviate concerns about our experiments.
>
> |Dataset|Network|Basic AT Framework| Method|Natural (Best)| PGD-20 (Best)| AA (Best)|Natural (Last)| PGD-20 (Last)| AA (Last)|
> | :-----| :-----| :-----|:-----| ----: | ----: |----: | ----: |----: | :----: |
> |CIFAR10|PreAct ResNet 18|AT|  $\mathrm{AT}$ |82.02 | 52.59| 48.23| 83.89| 45.28| 42.88|
> |CIFAR10|PreAct ResNet 18|AT| $\mathrm{SAAT_{up}}$ |76.37 | **56.31**| **48.86**| 76.38| **53.86**| **47.17**|
> |CIFAR10|PreAct ResNet 18|AT| $\mathrm{AWP-AT}$ |81.47 | 55.54| 49.96| 80.20| 54.88| 49.28|
> |CIFAR10|PreAct ResNet 18|AT| $\mathrm{AWP-SAAT_{up}}$ |79.49 | **56.25**| **50.67**| 77.91| **55.22**| **49.29**|
> |CIFAR100|PreAct ResNet 18|AT| $\mathrm{AT}$ |55.95 | 28.84| 24.71| 56.75| 21.12| 19.35|
> |CIFAR100|PreAct ResNet 18|AT| $\mathrm{SAAT_{up}}$ |53.20 | **29.29**| **25.25**| 54.52| **22.32**| **20.12**|
> |CIFAR100|PreAct ResNet 18|AT| $\mathrm{AWP-AT}$ |56.64 | 32.02| 26.43| 57.18| 31.64| 26.45|
> |CIFAR100|PreAct ResNet 18|AT| $\mathrm{AWP-SAAT_{up}}$ |54.36 | **32.09**| **26.98**| 55.08| **31.72**| **27.04**|
> |CIFAR10|PreAct ResNet 18|TRADES| $\mathrm{TRADES}$ |81.09 | 52.70| 48.91| 82.40| 49.98| 47.02|
> |CIFAR10|PreAct ResNet 18|TRADES| $\mathrm{SATRADES_{up}}$ |81.76 | **53.44**| **50.04**| 81.30| **51.13**| **47.99**|
> |CIFAR10|PreAct ResNet 18|TRADES| $\mathrm{AWP-TRADES}$ |81.64 | 55.52| 51.38| 81.87| 55.41| 51.27|
> |CIFAR10|PreAct ResNet 18|TRADES| $\mathrm{AWP-SATRADES_{up}}$ |80.76 | **55.98**| **51.72**| 80.64| **55.62**| **51.60**|
> |CIFAR10|Wide ResNet-34-10|AT| $\mathrm{AT}$ |85.47 | 54.87| 51.70| 86.16| 46.94| 45.68|
> |CIFAR10|Wide ResNet-34-10|AT| $\mathrm{SAAT_{up}}$ |79.94 | **58.52**| **52.36**| 80.19| **55.47**| **49.98**|
> |CIFAR10|Wide ResNet-34-10|AT| $\mathrm{AWP-AT}$ |86.00 | 58.78| 54.06| 87.08| 57.96| 53.41|
> |CIFAR10|Wide ResNet-34-10|AT| $\mathrm{AWP-SAAT_{up}}$ |83.55 | **59.20**| **54.64**| 84.05| **58.76**| **54.16**|

---

### Official Review · Reviewer_HbG6 · 2022-11-03

**Confidence:** 4
**Correctness:** 3
**Technical Novelty And Significance:** 3
**Empirical Novelty And Significance:** 3
**Recommendation:** 3

**Clarity, Quality, Novelty And Reproducibility:**

Clarity:

- The paper is clearly written and easy to follow.

Quality:

- The paper is well written but to support the claim more experiments need to be done.

Novelty:

- The idea is novel and interesting. As far as I know, no one has proposed similar adaptive ideas, but there are some works about adaptive epsilon in adversarial training.

Reproducibility:

- Though some implementation details are mentioned in the paper, code is not provided.

**Strength And Weaknesses:**

Strength:

- The paper is clearly written with intuition and finding discussed in detail.

- Experiments show that the proposed method outperforms standard adversarial training on CIFAR10

Weakness:

- The proposed method lacks theoretical and empirical support. Even though, some experiments are done on CIFAR10, but one dataset is not enough to show that the method works well in general. Besides, there have been many new adversarial training methods proposed since 2018 (the year standard adversarial training was proposed). The author also discusses some in the paper, but why not compare with those methods in the experiments? There have also been works about adaptive adversarial training. Lack of experiments is the key issue of the paper.

**Summary Of The Paper:**

The paper discusses the issue of adversarial training over-fitting, pointing out that a pre-specified perturbation budget is not optimal as the training progresses the perturbation budget should be adjusted accordingly. Based on this intuition, the author proposes a new adversarial training method that generates adversarial examples by maintaining a minimum adversarial loss instead of searching for the point that maximizes the loss. Experiments on CIFAR10 show that the proposed method outperforms standard adversarial training and when combined with AWP, it also outperforms standard adversarial training with AWP.

**Summary Of The Review:**

Due to the lack of experiments support, the paper is below the acceptance threshold.

---

> ### Author Response · Authors · 2022-11-13
> **Response to Reviewer HbG6**
>
> Thanks for your careful and valuable comments. We will explain your concerns point by point.
>
> **Q:** Lack of experiments.
>
> **A:** There have been many new adversarial training methods proposed since 2018, but we have yet to see work on adaptive adversarial budgets that can outperform standard AT in adversarial robustness (under the commonly used adversarial budget $\epsilon = 8/255$). IAAT[1] and MMA[2] are two adaptive adversarial budget methods, where IAAT improves natural accuracy but degrades adversarial robustness. MMA also degrades adversarial robustness (compared to Standard AT, see #14 of Table 1 in [3]). Thus, comparing with Standard AT and AWP can verify the effectiveness of our proposed method. On the other hand, due to the overestimated robustness issue of PGD, many methods can be easily broken by strong attacks (such as AA), whose robustness is actual lower than that of Standard AT, which has been mentioned in the paper. Under the AA's [evaluation criteria](https://github.com/fra31/auto-attack), Standard AT and AWP are still very competitive baselines. Then regarding the lack of experiments, we conducted related experiments across different datasets, different network structures and different AT frameworks, and the results are summarized in the following table. Experimental results show that the proposed method can achieve higher adversarial robustness under all settings. We will add these experimental results to the paper, hoping that these experimental results can alleviate concerns about the effectiveness and generality of the method.
>
> [1] Yogesh Balaji, Tom Goldstein, Judy Hoffman: Instance adaptive adversarial training: Improved accuracy tradeoffs in neural nets. CoRR abs/1910.08051 (2019)
>
> [2] Gavin Weiguang Ding, Yash Sharma, Kry Yik Chau Lui, Ruitong Huang: MMA Training: Direct Input Space Margin Maximization through Adversarial Training. ICLR 2020
>
> [3] Francesco Croce, Matthias Hein: Reliable evaluation of adversarial robustness with an ensemble of diverse parameter-free attacks. ICML 2020
>
> |Dataset|Network|Basic AT Framework| Method|Natural (Best)| PGD-20 (Best)| AA (Best)|Natural (Last)| PGD-20 (Last)| AA (Last)|
> | :-----| :-----| :-----|:-----| ----: | ----: |----: | ----: |----: | :----: |
> |CIFAR10|PreAct ResNet 18|AT|  $\mathrm{AT}$ |82.02 | 52.59| 48.23| 83.89| 45.28| 42.88|
> |CIFAR10|PreAct ResNet 18|AT| $\mathrm{SAAT_{up}}$ |76.37 | **56.31**| **48.86**| 76.38| **53.86**| **47.17**|
> |CIFAR10|PreAct ResNet 18|AT| $\mathrm{AWP-AT}$ |81.47 | 55.54| 49.96| 80.20| 54.88| 49.28|
> |CIFAR10|PreAct ResNet 18|AT| $\mathrm{AWP-SAAT_{up}}$ |79.49 | **56.25**| **50.67**| 77.91| **55.22**| **49.29**|
> |CIFAR100|PreAct ResNet 18|AT| $\mathrm{AT}$ |55.95 | 28.84| 24.71| 56.75| 21.12| 19.35|
> |CIFAR100|PreAct ResNet 18|AT| $\mathrm{SAAT_{up}}$ |53.20 | **29.29**| **25.25**| 54.52| **22.32**| **20.12**|
> |CIFAR100|PreAct ResNet 18|AT| $\mathrm{AWP-AT}$ |56.64 | 32.02| 26.43| 57.18| 31.64| 26.45|
> |CIFAR100|PreAct ResNet 18|AT| $\mathrm{AWP-SAAT_{up}}$ |54.36 | **32.09**| **26.98**| 55.08| **31.72**| **27.04**|
> |CIFAR10|PreAct ResNet 18|TRADES| $\mathrm{TRADES}$ |81.09 | 52.70| 48.91| 82.40| 49.98| 47.02|
> |CIFAR10|PreAct ResNet 18|TRADES| $\mathrm{SATRADES_{up}}$ |81.76 | **53.44**| **50.04**| 81.30| **51.13**| **47.99**|
> |CIFAR10|PreAct ResNet 18|TRADES| $\mathrm{AWP-TRADES}$ |81.64 | 55.52| 51.38| 81.87| 55.41| 51.27|
> |CIFAR10|PreAct ResNet 18|TRADES| $\mathrm{AWP-SATRADES_{up}}$ |80.76 | **55.98**| **51.72**| 80.64| **55.62**| **51.60**|
> |CIFAR10|Wide ResNet-34-10|AT| $\mathrm{AT}$ |85.47 | 54.87| 51.70| 86.16| 46.94| 45.68|
> |CIFAR10|Wide ResNet-34-10|AT| $\mathrm{SAAT_{up}}$ |79.94 | **58.52**| **52.36**| 80.19| **55.47**| **49.98**|
> |CIFAR10|Wide ResNet-34-10|AT| $\mathrm{AWP-AT}$ |86.00 | 58.78| 54.06| 87.08| 57.96| 53.41|
> |CIFAR10|Wide ResNet-34-10|AT| $\mathrm{AWP-SAAT_{up}}$ |83.55 | **59.20**| **54.64**| 84.05| **58.76**| **54.16**|
>
> The above is our explanation. If you have anything unclear or have further comments, please feel free to give us feedback and look forward to discussing with you. Thanks again for your time.

---

> > ### Author Response · Authors · 2022-11-15
> > **Reminder - Discussion Stage 1 closing soon - 18 November**
> >
> > We much appreciate the time and effort that you have dedicated to reviewing our manuscript. Just a reminder that we have explained your concerns about our manuscript, and please let us know if you have any questions about these explanations or new comments. We look forward to your feedback.

---

> > ### Author Response · Authors · 2022-11-17
> > **Final Reminder - Discussion Stage 1 closing soon - 18 November**
> >
> > A final reminder that discussion stage 1 is closing soon. Please let us know if you have further questions about our explanations, and thanks again for the time and effort that you have dedicated to reviewing our manuscript.

---

### Author Response · Authors · 2022-11-18
**Summary of Revision**

Dear Reviewers,

We greatly appreciate the time and effort that you have dedicated to reviewing our manuscript. We have revised our manuscript in light of your suggestions and the main changes are summarized here:

1.The performance evaluation across different datasets, different network structures and different AT methods is provided in Table 2 and Table 3 as suggested by all reviewers.\
2.Replace line 12 of Algorithm 1 with return x' as suggested by reviewer snup.\
3.Removed the description of "Further research" in Section 5 (Section Conclusion), as it has already been done.

By the way, we would like to highlight the contribution of our work. In addition to the robustness disparity proposed and defined in our work, and the proposed scheme for the robustness disparity problem, to our best knowledge, our method is also the first adaptive budget-based method to achieve robustness better than Standard AT. Moreover, our method provides a new perspective to alleviate robust overfitting and achieve performance improvements complementary to AWP, which we believe can bring some new knowledge to the community, especially for the in-depth exploration of the mechanism of robust overfitting.

Sincerely,\
The authors

---

### Author Response · Authors · 2022-11-24
**Message to meta reviewer**

Dear meta reviewer,

We hope you are all great. Although we have received negative comments, we have tried our best to address them. We are hoping to discuss with the reviewers during the rolling discussion phase to improve the paper. Can you kindly help coordinate and ask the reviewers to at least acknowledge reading our rebuttal? Thanks very much.

Best wishes,
Authors.

---

### Decision · Program_Chairs · 2023-01-20

**Decision:**

Reject

**Justification For Why Not Higher Score:**

N/A

**Justification For Why Not Lower Score:**

N/A

**Metareview: Summary, Strengths And Weaknesses:**

This paper proposes Strength-Adaptive Adversarial Training (SAAT) algorithm, where the perturbation budget will be adaptively adjusted according to the training state of adversarial data and can avoid robust overfitting.

Strengths:

+ The paper is well-written.

Weaknesses:

- The idea of adaptive attack budget has been adopted in prior works.

- The proposed algorithm lacks theoretical justification.

Even after the author response, the concerns remain unresolved. Therefore, I have to recommend rejection.